# Chemical Analysis of Minor Bioactive Components and Cannabidiolic Acid in Commercial Hemp Seed Oil

**DOI:** 10.3390/molecules25163710

**Published:** 2020-08-14

**Authors:** Luana Izzo, Severina Pacifico, Simona Piccolella, Luigi Castaldo, Alfonso Narváez, Michela Grosso, Alberto Ritieni

**Affiliations:** 1Department of Pharmacy, University of Naples “Federico II”, 80131 Naples, Italy; luigi.castaldo2@unina.it (L.C.); alfonso.narvaezsimon@unina.it (A.N.); alberto.ritieni@unina.it (A.R.); 2Department of Environmental, Biological and Pharmaceutical Sciences and Technologies, University of Campania “Luigi Vanvitelli”, 81100 Caserta, Italy; severina.pacifico@unicampania.it (S.P.); simona.piccolella@unicampania.it (S.P.); 3Department of Molecular Medicine and Medical Biotechnology, School of Medicine, University of Naples “Federico II”, CEINGE-Biotecnologie Avanzate, 80131 Naples, Italy; michela.grosso@unina.it; 4Health Education and Sustainable Development, “Federico II” University, 80131 Naples, Italy

**Keywords:** hemp seed oil, bioactive compounds, polyphenols, ω-6/ω-3 ratio, tocopherols, chlorophylls, carotenoids, cannabidiolic acid

## Abstract

Although hemp seed (HS) oil is characterized by more than 80% polyunsaturated fatty acids (PUFAs), a very high ω-6-to-ω-3 ratio is not a popular commodity. The aim of this work was to provide useful data about the bioactive components and cannabidiolic acid content in thirteen different commercial hemp seed oils. The investigated HS oils showed a good ω-6/ω-3 ratio, ranging from 1.71 to 2.27, massively differed in their chlorophylls (0.041–2.64 µg/g) and carotenoids contents (0.29–1.73 µg/g), as well as in total phenols (22.1–160.8 mg Gallic Acid Equivalents (GAE)/g) and tocopherols (3.47–13.25 mg/100 g). Since the high content of PUFAs in HS oils, photo-oxidative stability was investigated by determining the Thiobarbituric Acid Reactive Substances (TBARS) assay and extinction coefficient K232 and K270 after the photo-oxidative test. The percentage of increase in K232 and K270 ranged from 1.2 to 8.5% and from 3.7 to 26.0%, respectively, indicating good oxidative stability, but TBARS showed a 1.5- to 2.5-fold increase in oxidative behavior when compared to the initial values. Therefore, the diversity in bioactive compounds in HS oils, and their high nutritional value, suggest the need for a disciplinary booklet that well defines agronomic and post-harvest management conditions for achieving a good food objective.

## 1. Introduction

Industrial hemp (*Cannabis sativa* L.) is an annual herbaceous plant original from Central Asia but is widely spread around different geographical zones through its ability to adapt and respond to climate changes [1]. Consequently, this plant can be found in many different forms according to the resultant genetic variability. It has been long used since ancient times with medicinal and nutritional purposes, among others, due to its rich chemical content [2]. More than 500 different compounds have been characterized in *C. sativa*, with phytocannabinoids, flavonoids, terpenoids and fatty acids being the most relevant families [3]. Nevertheless, those compounds are not equally distributed throughout the plant, so the chemical content can drastically vary [4].

Currently, the EU Approved Common Catalogue of Cultivars includes some *C. sativa* cultivars with a total content of ∆^9^-tetrahydrocannabinol (∆^9^-THC) below 0.2% [5]. On the other hand, these cannabis cultivars popularly called “industrial hemp” generally contain a high concentration of the acidic precursor of cannabidiol (CBDA) [6,7]. The latter compound is known to have a wide range of important biological properties including anticonvulsive, anti-epileptic, and antimicrobial activities, and is also used as supplements in the treatment of osteoarthritis and for musculoskeletal disease [8].

In this line, the seed of hemp stands as a great source of nutrients and no-nutrients, containing 25–35% lipid, 20–25% protein and 30% carbohydrates [9]. Nowadays, the use of hemp for nutrition purposes is focused on the oilseed, mainly due to a rich content of polyunsaturated fatty acids (PUFAs) and a high ω-6 to ω-3 ratio of fatty acids, that reaches a 3:1 proportion. This value is considered as optimal for human health, with a positive influence in health condition by reducing the risk of suffering from cardiovascular diseases [10].

However, the high content of PUFAs in hemp seed oil makes it highly susceptible to lipid oxidation [11]. Oxidative instability is one of the most important factors responsible for reducing oil quality and shelf life [12]. This process in edible oils affects nutrition, toxicity, color and aroma, leading to the development of various off-flavors and an unpleasant taste, important reasons for consumers’ rejection [13]. Furthermore, it has been reported that the hemp seed oil contains a huge amount of chlorophylls that are able to interfere with oxidative stability and rancidity [14]. These natural pigments act as powerful prooxidants, increasing the susceptibility to light-induced oxidation or photo-oxidation of the edible oils when exposed to light and promote change from the intensive dark green color to yellow [15,16]. Several refining processes are needed during the hemp seed oil production to reduce the chlorophyll content and other minor components such as metals and phospholipids that could affect the oil quality.

Moreover, the hemp seed oil contains other minor constituents such as polyphenols, carotenoids, and tocopherols, all involved in antioxidant processes, which could play an important role in the protection of edible oils against lipid oxidation [17,18,19]. In humans, all these compounds can display important biological properties such as antioxidant and anti-inflammatory effects [20,21,22]. Because of this, hemp seed oil could become an alternative for other oil typologies in current diets, and would also be suitable for vegetarian, vegan or gluten-free diets even more considering their uprising trends. Although several hemp-based oily products are commercially available, the legal frame for marketing this kind of product in the European Union is still ambiguous. In fact, despite being classified as a novel food, no quality controls are established for marketed hemp seed oil, whereas olive oil has to fit several quality requirements set by the authorities [23]. Acidity, peroxide index, absorbance characteristics through K232 and K270 measurements and chemical composition, are the parameters that commonly qualify an olive oil as suitable for marketing, but their applicative definition for hemp seed oils could be limiting. Hemp seed oil quality should be guaranteed by more restrictive parameters that take into account its chemical complexity and its differentiation by means of several factors such as genetic factors (variety), or methods used for its obtainment (e.g., pressing and solvent extraction), as well as the refining and bleaching processes [24]. The development of shared standards to improve hemp seed oil quality management at a national level should be deeply pursued through the definition of a special disciplinary booklet. Although some studies have examined the chemical composition of hemp seed and hemp seed oils, little is known about the characteristics of commercial hemp seed oil that actually reaches the consumers [6,25,26]. Therefore, this study aims to provide useful data regarding the chemical composition as well as quality parameters of thirteen commercial hemp seed oils available in the Italian market in order to evaluate their biochemical characteristic variability and propose their introduction in the habitual human diet. The oil oxidation stability, evaluated by measuring the primary and secondary oxidation products during accelerated photo-oxidation tests, was also investigated.

## 2. Results and Discussion

Thirteen commercial hemp seed (HS) oils, which were from different industrial hemp varieties cultivated in Italy (Figure 1A), and produced as monovarietal oil or blended oil (Figure 1B), were investigated. HS oils, all obtained by cold-pressing extraction, are from different geographical origins, and are characterized by a large color variability (from light yellow to dark green) (Figure 1C), which could be related to a diverse content of chlorophylls, and other pigments such as carotenoids and polyphenols. The content of these latter compounds, together with that of other antioxidants in hemp seeds such as tocopherols, was valued. Furthermore, ω-6/ω-3 fatty acids ratio was assessed, and different conditions and parameters allowed oxidative stability to be assessed. Moreover, cannabidiolic acid amount was estimated in each analyzed hemp seed oil.

### 2.1. Polyphenols Determination of Hempseed Oil

Recently, hydroxycinnamoyl amides, lignanamides, and flavonol glycosides were identified as minor constituents of hemp seed [18,27], and their presence is favorably revealed in hemp seed meal, whereas their content is highly variable in hemp seed oil, being strongly dependent on the extraction method applied, and/or cultivar considered for oil obtainment [28]. Smeriglio et al. [29] also characterized the polyphenolic compounds of cold-pressed seed oil from Finola cultivar, finding a Total Phenol Content (TPC) equal to 267.5 mg Gallic Acid Equivalents (GAE)/100 g. The cultivar identity could also affect TPC content, as a value 10-fold lower (2.1 mg GAE/100 g) was estimated for hemp seed oil from Fedora cultivar by Siano et al. [26], who monitored the distribution of hemp seed components in the flour and oil after fractionation by cold-pressing. In the study conducted by Yu et al. [30] who evaluated the bioactivity of cold-pressed hemp seed oils for their potential application in the prevention of oxidative stress related diseases, a content of 44 mg GAE/100 g was reported. Moreover, Teh et al. [31] who studied the physicochemical and quality characteristics of cold-pressed New Zealand hemp seed oil, reported a TPC of 188.23 mg GAE/100 g. However, while considering the high variability in TPC values, and that total polyphenols by Folin–Ciocalteu (FC) method might be biased by several interfering components, including sugars and free amino acids [26], FC assay is rapid, applicable in routine laboratory use, and advantageously utilizable for screening antioxidant activity. This latter is the most known property attributed to phenolic compounds, which are also responsible for other relevant effects including nutritional characteristics, stability and the preventive effect against quenching radical reactions. Thus, investigated hemp seeds oils were properly fractionated, and total phenols content was determined on polyphenols enriched fractions obtained by means of FC method. Data acquired highlighted that TPC ranged from 36.1 to 160.8 mg GAE/g (Table 1), and strong differences were found taking into account oils obtained also from the same cultivar, cultivated and harvested in different geographical areas. This is the case of USO 31 cv. -based oils **1** and **9**, as well as Futura 75 cv. oils **4** and **13**. In particular, it was observed that TPC was 1.47-fold higher in oil **1** than in oil **9**, whereas Futura 75 cv. oil **13** contained more than double TPC in respect to oil **4**. The results suggest that, beyond the cultivar, geographical localization could massively affect total phenol content, whereas oils from the innermost areas show a lower content. Furthermore, drying methods (artificial or natural), and drought stress could benefit polyphenols enrichment. This is particularly true considering samples **4** and **13**, the first of which was from irrigated fields-derived hemp seeds, which were then naturally dried. According to Leonard et al. [32], hemp seeds’ total phenolic content varies depending on the cultivar, and within the seed itself differs from one fraction to another. Based on seed growth location, seed maturity, extraction conditions, resulting oil could have a strong and pungent flavor. Postharvest management practices could also affect compounds formation and consequentially the taste. It seems that cold-pressed hemp seed has the best quality almost compared to walnuts and sunflower seeds [33].

### 2.2. Tocopherols Determination in Investigated Hemp Seed Oils

Hemp seeds count in their antioxidant baggage vitamin E vitamers such as tocopherols, and tocotrienols, which are minor components of the hempseed oily fraction. These compounds are known to preserve the oxidative stability of oils acting as chain-breakers, and slowing-down lipo-peroxidation. Thus, these constituents are able to prevent the oxidation of PUFAs-rich oils [28] positively affecting their storage [34]. Among vitamin E vitamers, γ-Tocopherol is found to be the most abundant, followed by α-, and δ-tocopherols [35]. The assessment of tocopherols quantification in the different hemp seed oils is carried data, and data are reported in Table 2. Total tocopherols content is in the range of 3.47–13.25 mg/100 g. These results are in accordance with Smeriglio et al. [29] who reported a total tocopherol content in hemp oil after cold-pressing of Finola seed corresponding to 11.40 mg/100 g. In another study conducted by Anwar et al. [36] a detailed analysis of hemp seed oil native to three agro-ecological zones in Pakistan was carried out. The reported content of total tocopherol was in the range of 63.03–85 mg/100 g. Similar results were obtained by The et al. [31] who found a total tocopherol value of 59.16 mg/100 g. Moreover, Aladić et al. [37] studied the various processes in supercritical conditions in order to determine fatty acids, tocopherols and pigment content. Results highlighted an amount of α-tocopherol ranging from 3.71 to 11.06 mg/100 g and a quantity of γ-tocopherol content 2–3 folds higher, correlated on applied process conditions. Loss of tocopherols content in vegetable oils was attributed to similar factors to those which influence lipid oxidation, such as storage time, oxygen exposure and temperature [38].

### 2.3. Essential Polyunsaturated Fatty Acids Ratio

HS oil is a rare fount of nourishment for human diet, in particular for vegetarians, because of their unique reported ratio in ω-6/ω-3, most represented by linoleic acid (LA; 18:2, n-6) and α-linolenic acid (ALA; 18:3, n-3) [39]. The optimum ratio is approximately 2.5:1–3:1, also recommended for human nutrition [40]. Herein, the UHPLC analysis, combined with the quadrupole-time hybrid mass spectrometer (QTOF) high resolution, was utilized to detect and quantify fatty acids in analyzed oils. Taking into account relative areas of ω-6 and ω-3 fatty acids, their dietary ratio was calculated and data, which were in the range of 1.71–2.27, are listed in Figure 2. Several factors could affect fatty acids content and modify oil characters. Cultivars, climatic, farming and light conditions could strongly influence the ratio of ω6/ω3, in particular during seed development leading to the different content of acids in plants. Possessing high levels of unsaturated fatty acids protects the seeds from frost during the cold months of the year [36,41,42].

As reported by Abdollahi et al. [43] who studied cultivars of Fedora 17 and seeds of two native populations from Iran planted in three different regions, fatty acid composition is influenced by the origin of seeds. Their results confirm the variability on ratio ω6/ω3 reporting a value between 2.59 and 7.88.

Moreover, Devi et al. [39] investigated six different extraction processes for hemp oil comparing resulting physicochemical properties. The process that led to the best-suited optimum ω6/ω3 ratio was offered by Soxhlet treatment. Results on ratio ω6/ω3 of other treatments were in the range of 1.99–9.24.

Uston-Argon et al. [44] evaluated fatty acid compositions of cold-pressed hemp seed oil from Turkey. Results showed that linoleic/α-linolenic acid ratios were between 2.96–3.27 confirming the optimal level for a healthy daily diet.

In a study, conducted by Porto et al. [24] the optimization of the extraction process for obtaining high-quality hemp seed oil from Felina cultivar was performed. Soxhlet and SC-CO_2_ extraction processes were evaluated and no significant differences were found. Results on ratio ω-6/ω-3 were in the range of 3.25–3.31. Soxhlet extraction process is considered a prevalent method with relatively low cost and high oil extraction efficiency, which at the same time requires much more time to extract the oil and necessitates the use of hazardous chemical solvents [35].

However, Anwar et al. [36] evaluated hemp seed oil from three different agro-ecological zones of Pakistan. Results showed that the fatty acid composition of hemp seed oil native to Pakistan falls in the recommended nutritional ratio of 18:2 n-6 to 18:3 n-3 [40]. The wild hemp that grows over vast areas of the north of Pakistan appears to be a potentially valuable crop from which products with nutraceutical value could be derived. In fact, comparing the typical unsaturated fatty acid profiles of common food oils (rape, soybean, and linseed), hemp seed oil appears to have the major beneficial results according to Apostol et al. [45].

As evidenced by several numbers of studies above-reported, hemp seed oil fatty acid composition is characterized by an optimum ratio ω-6/ω-3. Polyunsaturated fatty acids are able to affect several biological activities and the optimal proportion ω-6/ω-3 of 3:1 has been claimed to have medicinal effects on the human body such as reducing cholesterol and high blood pressure, providing an anti-inflammatory effect and immune support effects, on the skin, diabetes and cardiovascular health [35,45].

### 2.4. Chlorophylls and Carotenoids in HS Oils under Study

The content of total chlorophylls and carotenoids in the different commercial hemp seed oil samples have been quantified as shown in Table 3. Total content of chlorophylls found in the different types of commercial hemp seed oil samples presented a wide range, from 0.41 (blend 2 and 3) up to 4.81 mg/kg (blend 1), with a mean value of 1.46 mg/kg for all samples. Our results are in accordance with data reported by Saastamoinen et al. [46] who analyzed the chemical composition of Finola hemp oil seeds cultivated in south-western part of Finland. Results showed a great difference in chlorophyll content among studied cultivars: 0–6.75 mg/kg. Instead, other studies have reported higher chlorophyll content in cold-pressed hemp seed oil up to 98.6 mg/Kg [47,48]. The natural chlorophyll content present in the samples analyzed here was effectively removed by conventional refining and bleaching processes in order to minimize the negative effects of high chlorophyll content in edible oils. Chlorophylls are undesirable, among minor components retained in cold-pressing oils. Their presence affects negatively the oils appearance with color from dark to light green, as well as oxidative stability. In fact, chlorophylls are photosensitizer and pro-oxidant species, able to induce PUFAs oxidation and to reduce, in turn, the hempseed oil shelf-life [49].

Carotenoids can be extracted along with chlorophylls by methanol, diethyl ether or other organic solvents, and determined by spectrophotometer at wavelength between 400 and 500 nm. The presence of carotenoids in hemp seed oil can protect chlorophylls from degradation and prevent any color change during storage [50]. Thus, to avoid photo-oxidation, minimizing chlorophylls’ content is mandatory. Furthermore, carotenoids were detected in a concentration range between 0.18 (blend 2) and 1.73 mg/kg (blend 1), with a mean value of 0.52 mg/kg for all samples. High levels of carotenoids have been previously reported in commercial hemp seed oils from Canada, at concentrations ten-fold greater when compared to the levels found in the assayed samples [51]. The wide difference in the content of the carotenoids in edible seed oils may be summarized in the variety and degree of seed maturity, climate characteristics during the plant growth, and the intensity of the bleaching process [52].

The ratio between the chlorophyll fraction and the carotenoid fraction greatly differed, demonstrating that the green and yellow fractions were not in balance (Figure 3). For other edible oils such as olive virgin oils, the ratio between the two isochromic fractions of pigments appeared to be constant at a value close to unity, independent of the variety [53]. There was a large variability for chlorophylls/carotenoids ratio within the hemp seed oil under study. In particular, hemp seed oil **8** was found to show the highest chlorophylls/carotenoids ratio, whereas only four samples, HS oils **4**, 7, **11**, and **13,** appeared to contain a good carotenoids amount and a ratio less than 2.5. Among these latter oils, although no conclusions are drawn, HS oils **4** and **13** are obtained from Futura 75 cv. seeds, and HS oil **11** is a blend in which 2/3 seeds were from Futura 75 variety. The chlorophylls/carotenoids ratio is able to indicate phenology and physiology of plants. Different varieties from different sources, harvested in a similar ripeness state, possess different chlorophyll and carotenoid pigment profiles and content. Different chlorophyll a/b content and chlorophyll/carotenoid ratios caused by stress, damage, and senescence, impact the normal course of plant biological processes. Even though there is not a restrict regulation on the ratio between chlorophylls and carotenoids, it tends to remain more or less constant whatever the variety of sources, in a concentration range of 2.5 to 3.7 mg total chlorophyll/mg total carotenoid [53].

### 2.5. Cannabidiolic Acid Content of Investigated Hemp Seed Oils

Cannabidiolic acid, the main phytocannabinoid compound in fiber and seed-oil plants, was detected and quantified by means of its HPLC-UV features [54]. Although the nutraceutical value of this compound is still far from being achieved, its high content in hemp varieties cultivated for food purposes, suggests the need for a deeper understanding of its detection and quantification, and to address its content assessment as a quality feature of hemp seed oil. Data acquired are presented in Figure 4. HS oil **5** showed the lowest CBDA content (0.28 mg per g of oil), where the blend 1 (HS oil **2**) was 14-fold more abundant. As for TPC value, it was observed that each oil could massively differ from the others according to the CBDA rate.

Our results are in accordance with Citti et al. [7] who analyzed cannabinoids contents in 13 different commercial Italian hemp seed oils. Results showed a concentration range between 2.265 and 233.8 mg/kg. Even though hemp seeds do not contain cannabinoids, their presence in them could be caused by direct contact with the resin secreted by the epidermal glands situated on flowers and leaves. So, cannabinoids represent “impurities” of the hemp seed oil probably deriving from the cleaning process of the seed. Moreover, the concentrations of cannabinoids can be highly changeable among different oil varieties.

To date, several countries including Austria, Finland, UK, Italy, and Germany allowed foods containing hemp products but lack or uncertainty regulation generating confusion in others. CBD food products may rely on Regulation (EU) 2015/2283 [55] in which CBD food products are considered as “Novel Food”. To comply with the new novel food guidelines, a concentration in CBD food products intended for internal use below 5% CBD is recommended.

### 2.6. Quality Parameters Determination

Acidity index, also indicated as acidity value, is an important indicator of vegetable oil quality and is expressed as the amount of potassium hydroxide (KOH) necessary to neutralize free fatty acids contained in 1 g of oil.

The A_I_ values for investigated oils (Table 4) are not in line with the maximum oleic acidity permitted for alimentary oils, which is estimated equal to 2%. Indeed, drying and storage are factors that could affect this value [56]. In particular, drying is reported to increase the index by 0.1%, whereas storage increases it by 0.05% per month. Taking into account that the main goal should be to commercialize hempseed oils with A_I_ values below 2%, with the only exception of HS oils **5** and **12**, the oils studied did not fully reach the desired objective. In fact, acidity index varied from 1.3 to 8.05%, but considering that Pharmacopoeia establishes a maximum value of 6% and 10% for hemp oil acidity and peroxide [57], most of the samples showed satisfactory results, with values fitting the above-mentioned requirements. In fact, when peroxide index, which is still the most common chemical method of measuring oxidative deterioration of oils, was evaluated, it was found in the range of 1.75–7.62 meq O_2_/kg for the analyzed samples.

As the formation of PUFAs hydroperoxide derivatives, which could occur in the early stages of an oxidation process, may result in the synthesis of conjugated dienic and trienic systems, spectrophotometric analyses were carried out to calculate Delta-k value (Table 4), which was less than 0.047 for all analyzed samples.

Quality parameters of hemp seed oil were evaluated by Jourdi et al. [58] who studied the physicochemical characterization of the fatty oils obtained by cold pressing of three hemp seeds harvested in ecological crops from Romania. Peroxide results were in the range of 0.6–1.2 meq O_2_/kg and acidity index was from 0.5 to 0.75%. In another study, [31] reported values for hemp seed oil of 1.94 meq O_2_/kg and 1.76% for peroxide and acidity index, respectively. As reported by Aladić et al. [37] who evaluated the extraction process from Croatian hemp seed oil by cold pressing, followed by extraction with supercritical CO_2_, methodology and the effects of temperature influenced quality parameters.

Although the quality parameters could not be compared with extra virgin oil [59], and the high acidity could impact on sensory characteristics of oil, such as flavor, the collateral characteristics make it still a very valuable oil. These oil benefits encourage the chemical industries versus the optimization of extraction process in order to accomplish the quality of the product [39].

### 2.7. Accelerated Photo-Oxidation Tests

Lipid oxidation is the main reaction responsible for reducing the nutritional value and shelf life of edible oils by the formation of oxidation products [60]. In accelerated photo-oxidation tests, the level of primary oxidation was evaluated by spectrophotometric absorbance at 232 nm which provides information on the conjugated dienes, and formation of products such as hydroperoxides [61]. Secondary lipid oxidation products such as aldehydes and ketones were monitored by absorbance of 270 nm, indicating the level of conjugated trienes as well as the content of carbonyl compounds [62]. Moreover, malondialdehyde (MDA) one of the main secondary oxidation products linked to the production of off-flavor and rancidity taste in edible oils was monitored by Thiobarbituric Acid Reactive Substances (TBARS) method [63]. Table 5 shows changes in K232- and K270-specific coefficients during the accelerated photo-oxidation tests. As far as the initial UV extinction coefficient was concerned, the K232 values of the commercial hemp seed oils were quantified at concentrations ranging from 0.81 (HS oil **10**-Felina 32) up to 0.86 units (HS oil **11**-Blend 4), with a mean value of 0.83 units for all samples. These levels were similar to the data previously reported in hemp seed oils [31,51]. On the other hand, the initial K270 monitored in different oil samples ranged from 0.34 (HS oil **5**-Blend 2) to 0.73 units (HS oil **2**-Blend 1), with a mean value of 0.51 units for all samples. All studied samples exceeded the limit of 0.25 units which has been established by the Commission Regulation (EC) for virgin olive oil [64].

After accelerated photo-oxidation tests, the percentage of increase in K232 and K270 ranged from 1.2 to 8.5%, and from 3.7 to 26.0%, respectively. In particular, the percentage of increase in K232 was significantly increased compared to initial oils samples (*p* < 0.05) only in HS oil 12. However, a significant increase in K270-specific coefficient during the accelerated photo-oxidation test was observed in four samples (Table 5). The results of this study indicated that the commercial hemp seed oils were resistant to photo-oxidation, which may be due to the presence of bioactive compounds such as polyphenols and carotenoids that play an important role in the oxidative stability [65].

On the other hand, the secondary oxidation of oils was also determined by the TBARs assay, as shown in Figure 5. The initial TBARS value was quantified at a concentration ranging from 0.35 up to 0.44 mmol MDA/kg. Concerning the occurrence of MDA in the oils after accelerated photo-oxidation tests, the levels found in assayed samples showed a 1.5- to 2.5-fold increase when compared to the initial values. Photo-oxidation susceptibility of the hemp seed oils may be due to the high content in unsaturated fatty acids recognized as vulnerable by oxygen, especially α-linolenic acid (ALA), that contributes the most to the degree of lipid peroxidation [66]. Pigments present in edible oil are known to play an important role in oxidative stability [67]. A weak linear relationship existed between the carotenoid levels and increased Thiobarbituric Acid Reactive Substances (TBARS) values after the accelerated photo-oxidation test (R^2^ = 0.43), but not with chlorophylls, suggesting that these pigments could provide some oxidative stability to oils. The significant increase in secondary products of lipid peroxidation showed after accelerated photo-oxidation tests limit the nutritive value as well as customer satisfaction [68]. Therefore, in order to protect the beneficial lipids in hemp oil, it is significantly important to promote better oxidative stability of hemp seed oils by the use of the dark opaque package, improving the lipid stability adding antioxidant compounds or develop a suitable formulation to encapsulate the hemp oil.

## 3. Materials and Methods

### 3.1. Reagents and Materials

All solvents, water (LC-MS grade), chloroform, methanol, *n*-hexane, iso-octane (2,2,4 trimethylpentane), and diethyl ether were acquired from Carlo Erba reagents (Milan, Italy), whereas ethanol absolute (≥99.8%), hydrochloric acid and formic acid (mass spectrometry grade) were purchased from VWR Chemicals (Milan, Italy).

Trichloroacetic acid (TCA, C_2_HCl_3_O_2_), 2-thiobarbituric acid (TBA, C_4_H_4_N_2_O_2_S), sodium thiosulphate (Na_2_S_2_O_3_), sodium carbonate (Na_2_CO_3_), potassium iodide (KI), Folin–Ciocalteu reagent, gallic acid, anhydrous sodium sulfate (Na_2_SO_4_) anhydrous sodium sulfate (Na_2_SO_4_) and phenolphthalein solution were obtained from Sigma Aldrich (Milan, Italy).

All standards (purity > 98%), namely malondialdehyde (MDA), butylated hydroxytoluene (BHT), gallic acid, α-tocopherol, oleic acid and α-tocopherol were acquired from Sigma Aldrich (Milan, Italy). Chemicals and reagents were of analytical grade.

### 3.2. Sampling

A total of thirteen samples of commercial hemp (*Cannabis sativa* L.) seed oil of different origins (Center and Southern Italy) were purchased from the Italian local market. Eight samples were monovarietal, including Codimono (n = 1), Felina 32 (n = 1), Futura 75 (n = 2), Secuieni jubileu (n = 1), Uso – 31 (n = 2) and Zenit (n = 1), whereas five oil samples were the result of a blend composed of several varieties such as Blend 1 (Carmagnola, Futura 75, KC Dora, Kompolti and Tibor); Blend 2 (Futura 75, Finola and Felina 32), Blend 3 (Uso-31, Secuieni jubileu and Zenit); Blend 4 (Futura 75 and USO-31); and Blend 5 (Uso-31, Futura 75, Felina 32 and Zenit). The percentages of each variety were not declared in labels from samples. All the samples were stored in dark and cool conditions until further analysis.

### 3.3. Polyphenols and Fatty Acids Extraction

Polyphenols extraction was performed in accordance with the procedure reported by Moccia et al. [19], with modifications. Briefly, 5 g of oil, previously dissolved in 5 mL of *n*-hexane, were extracted with 50 mL of MeOH:Me_2_CO:H_2_O (7:7:6, *v:v:v*) solution. The mixture was intensively vortexed for 3 min and sonicated for 30 min in an ultrasonic bath. Then, the mixture was centrifuged 5000× *g* at 4 °C for 10 min. The lower phases from three consecutive extractions were combined, filtered through 0.45 mm vacuum membranes (Millipore, Billerica, MA, USA) and concentrated using rotary evaporation in a water bath at 30 °C, and analyzed for their total phenolic content (please see Section 3.4). The upper phases were washed three times with MeOH:H_2_O (4:1, *v*:*v*) solution to remove non-lipid substances, anhydrified on Na_2_SO_4_ and then investigated for their polyunsaturated fatty acids (please see Section 3.6).

### 3.4. Determination of Total Phenolic Content (TPC)

Total phenolic content was determined based on Folin Ciocalteu method [69]. In short, 0.125 mL of extract was diluted in 0.5 mL of deionized water and reacted with 0.125 mL of Folin–Ciocalteu reagent for 6 min in dark conditions at room temperature. Then, 1.25 mL of 7.5% of sodium carbonate solution and 1 mL of deionized water were added to reach a final volume of 3 mL. The absorbance at 760 nm after 90 min of incubation in the dark was determined using an UV/Vis Spectrophotometer (DU 730, Beckman Coulter, Brea, CA, USA). Results was expressed as mg of gallic acid equivalents GAE/g of sample.

### 3.5. α-Tocopherol Extraction and Determination

α-Tocopherol extraction was performed according to the method of Mallek-Ayadi et al. [70], with slight modifications. In short, 0.1 mL of oil was diluted in 0.9 mL of n-hexane contained in an Eppendorf tube. From this mixture, 0.2 mL was transferred in an Eppendorf tube in which 0.8 mL of methanol was present. The mixture was vortexed for 3 min and centrifuged at 1008× *g* for 5 min at 15 °C. The supernatant was filtered through 0.45 mm vacuum membranes (Millipore, Billerica, MA) and kept at 4 °C until HPLC analysis.

α-tocopherol was assessed by HPLC/diode-array detector (DAD) analysis, performed using an HPLC system Jasco Extrema LC-4000 system (Jasco Inc., Easton, MD, USA) fitted with an autosampler, a binary solvent pump, and a diode-array detector (DAD), according to Chen et al. 2010. The separation and quantification were carried out using Prodigy C18 column (250 × 4.6 mm, 5 µm particle size; Phenomenex, Castel Maggiore, Italy) preceded by a security guard cartridge. The column temperature was set at 30 °C. The photometric diode array (PDA) acquisition wavelength was set in the range between 200 and 400 nm. The mobile phase consisted of isocratic CH_3_OH:H_2_O (98:2 *v*/*v*) with a flow rate of 1 mL/min. The injection volume was set at 20 µL. α-tocopherol was identified by comparison with a pure standard (Sigma Chemicals, Milan, Italy); the calibration curve was obtained by measuring absorbance at 290 nm, over the concentration range of 0.5–500 mg/kg.

### 3.6. UHPLC-ESI-QqTOF-MS/MS Analysis for PUFAs Determination

Hexane fractions from liquid–liquid extraction (please see Section 3.3) were analyzed for PUFA content [28,71]. For this purpose, a Shimadzu NEXERA UHPLC system was used with a Luna^®^ Omega Polar C_18_ column (1.6 μm particle size, 150 × 2.1 mm i.d., Phenomenex, Torrance, CA, USA). Separation was achieved with a linear gradient of water (A) and acetonitrile (B), both with 0.1% formic acid. Gradient conditions were as follows: 0–5 min, linear from 5 to 55% B; 5–10 min, linear from 55 to 75% B; 10–11 min, from 75 to 95% B; 11–12 min, isocratic 95% B. Then, at 12.01 min the starting conditions were restored and the column was allowed to re-equilibrate for 2 min. The total run time was 14 min, with a flow rate of 0.5 mL/min and an injection volume of 2.0 μL. MS analysis was performed using a hybrid QqTOF MS instrument, the AB SCIEX TripleTOF^®^ 4600 (AB Sciex, Concord, ON, Canada), equipped with a DuoSprayTM ion source (consisting of both electrospray ionization (ESI) and atmospheric pressure chemical ionization (APCI) probes), which was operated in the negative ESI mode. The APCI probe was used for automated mass calibration using the Calibrant Delivery System (CDS). The CDS injects a calibration solution matching the polarity of ionization, and calibrates the mass axis of the TripleTOF^®^ system in all scan functions used (MS and/or MS/MS). The QqTOF HRMS method, which combines TOF-MS and MS/MS with Information Dependent Acquisition (IDA) for identifying non-targeted and unexpected compounds, consisted of a full scan TOF survey (dwell time 100 ms, 100–1000 Da) and a maximum number of eight IDA MS/MS scans (dwell time 50 ms, 80–85 Da)]. Quantitation is carried out taking into account peak area from selected ion chromatograms (three independent measurements were performed). In particular, the [M − H]^−^ ion for α-linolenic acid was at *m/z* 277.2173, according to the molecular formula C_18_H_30_O_2_ (calcd. 277.2176 error ppm 1.4, unsaturation degree 4) and TOF-MS/MS fragment ions were at *m/z* 259.2072 ([M−H-H_2_O]^−^), 233.2275 ([M − H-CO_2_]^−^). The [M − H]^−^ ion for linoleic acid was at *m/z* 279.2330, according to the molecular formula C_18_H_30_O_2_ (calcd. 279.2331 error ppm 0.5, unsaturation degree 3) and TOF-MS/MS fragment ion was at *m/z* 261.2225 ([M − H-H_2_O]^−^).

### 3.7. Oil Pigment Determination

Chlorophyll and carotenoid content of the hemp seed oil samples were assessed based on the method reported by Isabel Minguez-Mosquera et al. [72] with slight modifications. Briefly, each oil sample (0.100 ± 0.001 g) was placed in a tube and the volume was made up to 3 mL using absolute diethyl ether. The solution was vortexed thoroughly and sonicated for 1 min. The absorbance of the solution was measured in the wavelength range 250–750 nm by UV-1700 UV/Vis spectrophotometer (Shimadzu, Kyoto, Japan) against a blank. Oil pigment content (µg/mL) was calculated according to the formula:(1)$$\mathrm{Chlorophyll}\;a=9.93\times A_{663}-0.78\times A_{640}.\;$$
(2)$$\mathrm{Chlorophyll}\;b=17.60\times A_{640}-2.81\times A_{663}$$
(3)$$\mathrm{Chlorophyll}\;a+b=7.12\times A_{663}+16.80\times A_{640}$$
(4)$$\mathrm{Total}\;\mathrm{carotene}=\frac{\left(1000\;\times A_{470}\;-\;0.52\times Chl\;a-7.25\times Chl\;b\right)}{226}.\;$$

### 3.8. Cannabidiolic Content Determination

Cannabidiolic acid (CBDA) content was estimated by HPLC-UV-DAD analysis according to Formato et al. [54]. To this purpose, the HPLC 1260 INFINITY II system (Agilent, Santa Clara, CA, USA) was utilized, equipped with an Agilent G7129A autosampler, an Agilent GY115A DAD-UV-visible detector and a Quaternary pump Agilent G711A. The analysis was carried out using the Luna^®^ Phenyl-Hexyl column (150 × 2 mm, 3 µm). The mobile phase consisted of a binary solution A: 0.1% HCOOH in H_2_O, B: 0.1% HCOOH in CH_3_CN. A linear gradient was started at 55% B, held for 1.5 min, and linearly ramped to 95% B in 6.50 min. The mobile phase composition was maintained at 95% B for another 2 min, then returned to the starting conditions and allowed to re-equilibrate for 3 min. The total analysis time was 13.00 min. The analyses were carried out in three independent measurements and the results were expressed as mean values ± Standard Deation (SD).

### 3.9. Quality Parameters Determination

Quality parameters of oil were evaluated according to the procedures indicated in Commission regulation (EU) No. 2016/1227 [73], No. 2016/1784 [74], No. 2015/1833 [75], amending Regulation (EEC) No. 2568/91 [23].

#### 3.9.1. Determination of Free Fatty Acids

An aliquot of 2.5 g of hemp oil sample was dissolved in 100 mL of a mixture of diethyl ether and ethanol (1:1 *v/v*). Phenolphthalein solution (1% in ethanol) was used as indicator. The free fatty acids present were neutralized using potassium hydroxide solution (0.1 mol/L) until the indicator color changes and persists for at least 10 s. Results were expressed in % of oleic acid, and calculated with the following formula:(5)$$\%\;=\frac{V\times c\times M}{10\mathrm{xm}}.\;$$
where: V = the volume potassium hydroxide solution used in the titrated (mL); c = the exact concentration in mol/L of potassium hydroxide solution; M = 282 g/mol, the mar mass of oleic acid; m = the mass of the sample (g).

#### 3.9.2. Determination of Peroxide Value

An aliquot of 2 g of hemp oil sample was dissolved in 25 mL of a mixture of acetic acid: chloroform (3:2 *v/v*) stirred until complete solubilization. Then, 1 mL of saturated KI solution was added and the flask was immediately closed and stirred for 2 min in the dark, the necessary time for the oxidation of the iodide to iodine from part of hydroperoxides. After, 75 mL of distilled water and 1 mL of starch solution were added before the titration with thiosulphate solution 0.01 N until the color blue-violet disappeared. The peroxide value, expressed in meq O_2_/kg, is given by:(6)$$\mathrm{PV}=\frac{V\times T\times 1000}{m}$$
where: V = the volume of thiosulphate solution 0.01 N used in the titrated (mL); T = the exact concentration in mol/L of thiosulphate solution; m = the mass of the sample (g).

#### 3.9.3. Spectrophotometric Investigation in the Ultraviolet

An aliquot of 0.25 g of hemp oil sample, weighted into a 25-mL graduated flask, was dissolved in 25 mL iso-octane. After exhaustive shaking, the solution was read at the wavelengths of 232, 270, 264, 268 and 272 nm by using a spectrophotometer instrument. Quartz cuvettes were used for the described test. The variation of the absolute value of the extinction (λk) is given by:(7)$$\lambda k=\mathrm{Km}\frac{\left(K\lambda m-4+\;K\lambda m+4\right)}{2}$$
where Km is the specific extinction at the wavelength for maximum absorption at 268 nm in iso-octane.

### 3.10. Preparation of Samples for Accelerated Photo-Oxidation Tests

The photo-oxidation procedure proposed by Abuzaytoun et al. [76] was selected as a starting point and then slightly modified. A 15 mL PTFE tube was filled with the oil so that the headspace was ∼1% of the volume. The test tube was placed in a thermo block (70 cm length × 35 cm width × 25 cm height) with fluorescent radiation of 2650 lux and an internal temperature of 24 °C. After incubation of 7 days, the commercial hemp seed oil samples were recovered from the oven and kept at −20 °C until analysis.

#### 3.10.1. Determination of the Lipid Photostability

Lipid photo-oxidation of the commercial hemp seed oils obtained during the accelerated photo-oxidation tests and non- oxidized samples was assessed and compared by TBARS test and UV–spectrophotometric analyses.

#### 3.10.2. Thiobarbituric Acid Reactive Substances (TBARS) Determination

The determination of TBARS was carried out according to the procedure described by Maqsood et al. [77] with minor modifications. Thiobarbituric acid (TBA) reagent was prepared as follows: for reagent A, TBA (375.0 mg) and tannic acid (30.0 mg) were dissolved in hot water (30.0 mL), for reagent B, trichloroacetic acid (15 g) was dissolved in an aqueous hydrogen chloride solution (0.30 M, 70.0 mL). Then, reagent A was mixed with reagent B. Hemp seed oil sample (0.25 g) was emulsified with Tween-40 (15.6 mg) initially dissolved in Tris-HCl buffer (0.2 M, 2.0 mL, pH 7.4). After addition of the TBA reagent (1.0 mL), all test tubes were placed in a boiling water bath for 15 min. Then, 500.0 μL of *n*-butanol was added, and centrifuged at 252× *g* for 3 min. The absorbance of supernatant was measured at 532 nm. Inhibition of lipid peroxidation was recorded as percentage vs. a blank containing no test sample [62]. Results were expressed as nmol of MDA equivalents/kg sample, by interpolation from the MDA standard curve.

#### 3.10.3. UV-Pectrophotometric Analyses

The photo-oxidation was evaluated monitoring the spectrophotometric absorbance at 232 and 270 nm of the samples after accelerated photo-oxidation tests compared to their respective non-oxidized samples. The results were expressed as K270 and K232.

### 3.11. Statistical Analysis

All experiments were conducted in triplicate and the results expressed as the average values ± standard deviation (SD). The differences between average values were evaluated by using the Student *t*-test at a significance level of 0.05. Correlation coefficients between the different experimental data were determined using Pearson’s test. Statistical analysis was carried out using STATA 12 (STATACorp LP, College Station, TX, USA).

## 4. Conclusions

Nowadays, consumers are constantly looking for useful natural products to supplement the human diet in order to prevent or treat human diseases. Despite some reticence towards this matrix, hemp seed oil has been heavily studied in recent decades. So, several studies reported the beneficial effect derived from oil extracted from hemp seed. Indeed, although the use of hemp seed in Italy, as part of human diet as it is or as its by-product (e.g., oil), has been legislated since 2009 [78], little is known about its chemical composition and the variability of this. This can have several primary consequences, among which is the marketing of highly dissimilar products from an analytical-quantitative point of view. The uniqueness of the hemp seed oil products must be the subject/object of greater attention. Data herein reported showed that investigated HS oils markedly differ in the content of some bioactive compounds, suggesting the need for a disciplinary booklet that well defines agronomic and post-harvest management conditions for achieving a good food objective.

## Figures and Tables

**Figure 1 molecules-25-03710-f001:**
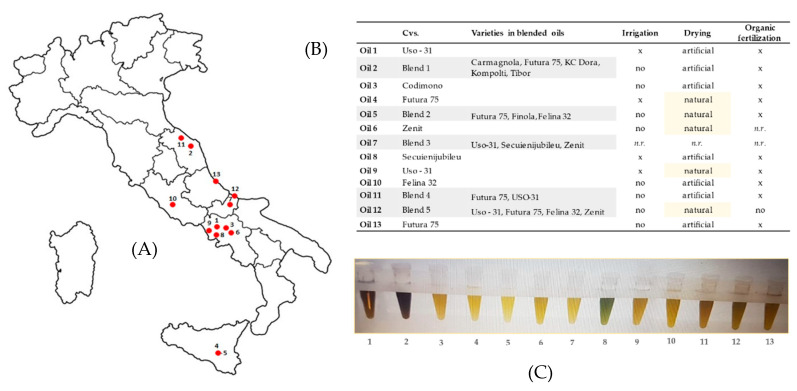
Cold-pressed hemp seed oils from different Italian locations. Main cultural and post-harvest techniques applied are indicated. *n.r. = not reported.* (**A**) Graphic distribution of hemp seed oils in the various Italian regions; (**B**) Main characteristics of hemp seed oils: cultivars, varieties, irrigation, drying and organic fertilization; (**C**) Visual representation of analyzed hemp seed oils.

**Figure 2 molecules-25-03710-f002:**
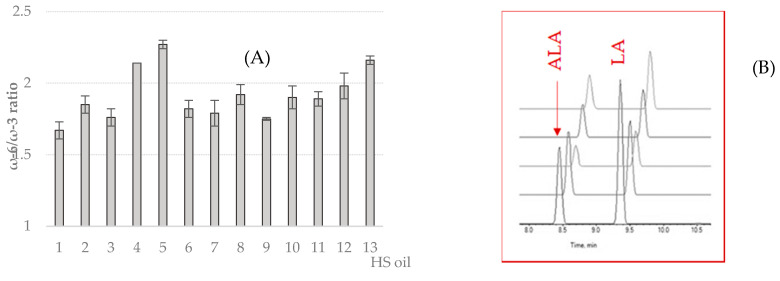
Fatty acids ratio ω-6/ω-3 evaluated on the different cultivars of hemp seed oils (**A**). Results are expressed as mean of three independent measurements ± SD. (**B**) A representative total ion current TIC’s enlarged part, underlining α-linolenic acid (ALA) and linolenic acid (LA) separation, is reported in the red square.

**Figure 3 molecules-25-03710-f003:**
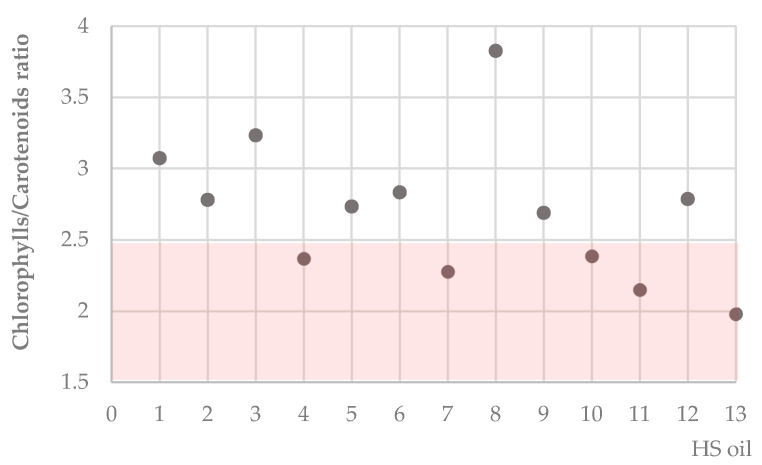
Chlorophylls/carotenoids ratio of the investigated hemp seed oils.

**Figure 4 molecules-25-03710-f004:**
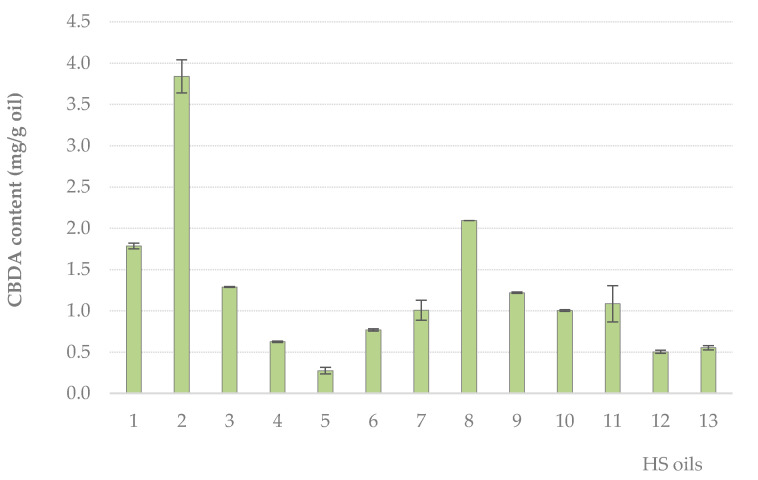
Cannabidiolic acid (CBDA) content (mg per g of oil) of the analyzed hemp seed oils, calculated by means of the calibration curve of a home-made isolated CBDA. Values are the mean of three independent measurements ± SD.

**Figure 5 molecules-25-03710-f005:**
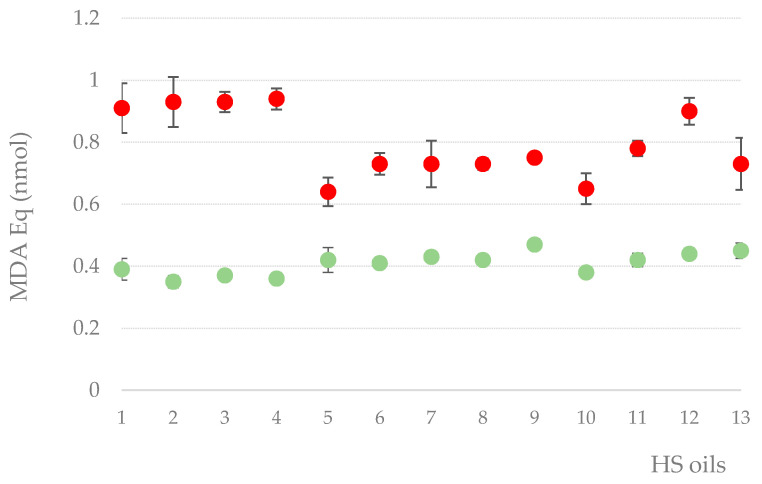
Thiobarbituric Acid Reactive Substances (TBARS) values, expressed as nmol of malondialdehyde (MDA) equivalents, in initial oils (●), and after 7 days accelerated photo-oxidation tests (●). Values are the mean of three independent measurements ± SD.

**Table 1 molecules-25-03710-t001:** Total Polyphenol Content (TPC) evaluated for hemp seed oil samples. Values are reported as mean ± SD of three independent measurements, and are expressed as mg Gallic Acid Equivalents (GAE) per g of oil.

Sample	Cvs.	TPC mg GAE/g
Oil 1	Uso-31	32.5 ± 4.9
Oil 2	Blend 1	160.8 ± 3.7
Oil 3	Codimono	48.9 ± 1.2
Oil 4	Futura 75	69.2 ± 8.6
Oil 5	Blend 2	58.2 ± 8.4
Oil 6	Zenit	36.7 ± 7.1
Oil 7	Blend 3	36.1 ± 2.3
Oil 8	Secuieni jubileu	108.0 ± 0.1
Oil 9	Uso-31	22.1 ± 0.3
Oil 10	Felina 32	124.3 ± 18.4
Oil 11	Blend 4	36.3 ± 4.8
Oil 12	Blend 5	105.0 ± 11.4
Oil 13	Futura 75	139.7 ± 14.8
Skewness		0.6
Kurtosis		−2.2

**Table 2 molecules-25-03710-t002:** Tocopherols determination evaluated for hemp seed oil samples. Results are expressed as mg 100/g ± SD from three independent measurements.

Sample	Variety	Tocopherols (mg/100 g)
Oil 1	Uso-31	7.23 ± 0.47
Oil 2	Blend 1	13.25 ± 0.18
Oil 3	Codimono	7.29 ± 0.20
Oil 4	Futura 75	6.55 ± 0.50
Oil 5	Blend 2	5.83 ± 0.34
Oil 6	Zenit	7.39 ± 0.02
Oil 7	Blend 3	6.46 ± 0.58
Oil 8	Secuieni jubileu	3.47 ± 0.32
Oil 9	Uso-31	6.25 ± 0.01
Oil 10	Felina 32	6.54 ± 0.22
Oil 11	Blend 4	7.42 ± 0.45
Oil 12	Blend 5	7.84 ± 0.20
Oil 13	Futura 75	8.23 ± 0.13
Skewness		1.6
Kurtosis		5.7

**Table 3 molecules-25-03710-t003:** Chlorophylls and carotenoids content in the analyzed oils (µg/g ± SD).

Sample	Variety	Chlorophylls *a + b* (µg/g ± SD)	Carotenoids (µg/g ± SD)
Oil 1	Uso-31	2.52 ± 0.27	0.82 ± 0.09
Oil 2	Blend 1	4.81 ± 0.12	1.73 ± 0.04
Oil 3	Codimono	0.97 ± 0.08	0.30 ± 0.02
Oil 4	Futura 75	0.45 ± 0.02	0.19 ± 0.00
Oil 5	Blend 2	0.41 ± 0.02	0.15 ± 0.01
Oil 6	Zenit	0.85 ± 0.01	0.30 ± 0.01
Oil 7	Blend 3	0.41 ± 0.06	0.18 ± 0.02
Oil 8	Secuieni jubileu	2.64 ± 0.76	0.69 ± 0.10
Oil 9	Uso-31	0.78 ± 0.03	0.29 ± 0.01
Oil 10	Felina 32	1.05 ± 0.07	0.44 ± 0.01
Oil 11	Blend 4	0.86 ± 0.08	0.40 ± 0.03
Oil 12	Blend 5	1.70 ± 0.19	0.61 ± 0.02
Oil 13	Futura 75	0.97 ± 0.01	0.49 ± 0.01
Skewness		1.9	2.3
Kurtosis		3.9	6.2

**Table 4 molecules-25-03710-t004:** Quality parameters in terms of acidity (%), peroxide index (meq O_2_/kg) and Delta-k spectrophotometric analysis measured for the hemp seed oils under study. Results are expressed as mean of three independent measurements ± SD.

Sample	Variety	Acidity (%)	Peroxide Value (meq O_2_/kg)	Delta-k
Oil 1	Uso-31	9.9 ± 0.08	6.5 ± 0.07	0.015± 0.003
Oil 2	Blend 1	7.9 ± 0.01	4.2 ± 0.01	0.029 ± 0.002
Oil 3	Codimono	5.1 ± 0.09	1.8 ± 0.15	0.022 ± 0.001
Oil 4	Futura 75	2.8 ± 0.06	6.8 ± 0.003	0.019 ± 0.001
Oil 5	Blend 2	1.3 ± 0.01	5.3 ± 0.003	0.013 ± 0.003
Oil 6	Zenit	5.1 ± 0.01	6.3 ± 0.75	0.019 ± 0.002
Oil 7	Blend 3	6.8 ± 0.08	4.8 ± 0.20	0.027 ± 0.001
Oil 8	Secuieni jubileu	4.9 ± 0.20	5.3 ± 0.50	0.029 ± 0.003
Oil 9	Uso-31	6.2 ± 0.03	8.8 ± 0.07	0.020 ± 0.002
Oil 10	Felina 32	2.1 ± 0.09	4.2 ± 0.30	0.047 ± 0.003
Oil 11	Blend 4	8.0 ± 0.07	3.8 ± 0.25	0.023 ± 0.001
Oil 12	Blend 5	1.7 ± 0.09	4.8 ± 0.15	0.024 ± 0.002
Oil 13	Futura 75	3.0 ± 0.06	3.0 ± 0.08	0.020 ± 0.003
Skewness		0.3	0.3	1.7
Kurtosis		−1.9	0.7	4.4

**Table 5 molecules-25-03710-t005:** Extinction coefficient at 232 and 270 nm in initial oils and after 7 days of accelerated photo-oxidation tests. Values are the mean of three independent measurements ± SD.

Sample	t_0_		t_7_	
	K232	K270	K232	K270
Oil 1	0.86 ± 0.08	0.69 ± 0.07	0.86 ± 0.03	0.79 ± 0.05
Oil 2	0.84 ± 0.07	0.73 ± 0.04	0.88 ± 0.04	0.81 ± 0.06
Oil 3	0.84 ± 0.03	0.51 ± 0.02	0.88 ± 0.04	0.54 ± 0.04
Oil 4	0.84 ± 0.01	0.39 ± 0.05	0.87 ± 0.02	0.42 ± 0.05
Oil 5	0.82 ± 0.08	0.34 ± 0.01	0.82 ± 0.03	0.38 ± 0.02 *
Oil 6	0.83 ± 0.03	0.40 ± 0.04	0.87 ± 0.02	0.44 ± 0.03
Oil 7	0.85 ± 0.04	0.54 ± 0.07	0.85 ± 0.03	0.56 ± 0.04
Oil 8	0.83 ± 0.03	0.65 ± 0.06	0.82 ± 0.04	0.70 ± 0.03
Oil 9	0.81 ± 0.02	0.46 ± 0.05	0.83 ± 0.03	0.47 ± 0.02
Oil 10	0.81 ± 0.05	0.50 ± 0.05	0.86 ± 0.03	0.63 ± 0.02 *
Oil 11	0.86 ± 0.02	0.50 ± 0.04	0.87 ± 0.02	0.60 ± 0.02 *
Oil 12	0.82 ± 0.01	0.47 ± 0.04	0.89 ± 0.04 *	0.54 ± 0.03
Oil 13	0.83 ± 0.08	0.36 ± 0.02	0.88 ± 0.07	0.43 ± 0.01 *

Statistical significance is calculated by Student’s t-test analysis: * *p* < 0.05 Extinction coefficient (232 or 270 nm) in initial oils vs. after 7 days of accelerated photo-oxidation tests.

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
