# Peer review of "Chemical Analysis of Minor Bioactive Components and Cannabidiolic Acid in Commercial Hemp Seed Oil"

_molecules, 2020, doi:10.3390/molecules25163710_

Round 1

Reviewer 1 Report

This study describes analysis of minor bioactive components and cannabidiolic acid in commercial hemp seed oil. Several quality parameters required for marketed oil were investigated as well. The topic is interesting, and extensive amount of experiments was conducted in this study. However, this study has some major issues to address: 1) I didn’t see novelty clearly through the entire manuscript. Authors should stress the novelty of this study (abstract and conclusion). 2) I am not sure how this work can be used in real world, although the data is abundant and informative. Author should discuss more about the applicability of this work. 3) More discussion will be needed for some parts of results (e.g. section 2.4, 2.5, PCA, etc.). My specific comments are listed below:

  1. Line 2-4: The title sounds like a review article, and does not fit the aim of this study (line 22-24).
  2. Abstract: There are only methodologies listed in Abstract. I couldn't see any meaningful results and discussion here. Moreover, novelty was not described clearly. I suggest that authors rewrite and reorganize Abstract.
  3. Line 87-90: This sentence is too broad and ambiguous. Please rephrase this sentence to make it more focused and clearer.
  4. Line 172-218: Can ratio of LA/ALA represent to ratio of total w-6/w-3 fatty acid? There must be some other w-3 and w-6 fatty acids in the hemp oil.
  5. Line 220-255: Please add some discussion for the results of chlorophylls and carotenoids (the meaning of their concentrations, ratios, global regulations, etc.).
  6. Line 251: What is chlorophylls/carotenoids ratio for? Any regulation for this?
  7. Line 260-268: Please add more discussion for the results of cannabidiolic acid (meaning, regulations, etc.).
  8. Line 359-370: What is the meaning of PCA analysis in this study? How can the results of PCA contribute to the quality control of hemp seed oil? I didn't get the point of grouping of oil samples on the PCA score scatter plot. The results need more discussion as well.
  9. Line 392: I am wondering if there were any biological and/or technical replicates for each sample?
  10. Line 459-460: Which method was used for LC-MS quantification (calibration curve, etc.)? On top of that, how did author correct deviations from extraction and MS ionization (internal standards, etc.)?

Author Response

Response to reviewers

Manuscript ID: molecules-883344

Title: Analyzing commercial hemp seed oil from Mediterranean basin: how far have we got in its chemical goodness?

Reviewer 1

Comments and Suggestions for Authors

This study describes analysis of minor bioactive components and cannabidiolic acid in commercial hemp seed oil. Several quality parameters required for marketed oil were investigated as well. The topic is interesting, and extensive amount of experiments was conducted in this study. However, this study has some major issues to address: 1) I didn’t see novelty clearly through the entire manuscript. Authors should stress the novelty of this study (abstract and conclusion). 2) I am not sure how this work can be used in real world, although the data is abundant and informative. Author should discuss more about the applicability of this work. 3) More discussion will be needed for some parts of results (e.g. section 2.4, 2.5, PCA, etc.). My specific comments are listed below:

  1. Line 2-4: The title sounds like a review article, and does not fit the aim of this study (line 22-24).

-As reviewer 1 suggested, the authors changed the title “Analyzing commercial hemp seed oil from Mediterranean basin: how far have we got in its chemical goodness?” to “Chemical analysis of minor bioactive components and cannabidiolic acid in commercial hemp seed oil”.

  1. Abstract: There are only methodologies listed in Abstract. I couldn't see any meaningful results and discussion here. Moreover, novelty was not described clearly. I suggest that authors rewrite and reorganize Abstract.

-As suggested by reviewer 1, the authors rewrite and reorganize abstract including the main results.

  1. Line 87-90: This sentence is too broad and ambiguous. Please rephrase this sentence to make it more focused and clearer.

-As suggested by reviewer 1, the authors rewrite the sentence in line 89-90.

  1. Line 172-218: Can ratio of LA/ALA represent to ratio of total w-6/w-3 fatty acid? There must be some other w-3 and w-6 fatty acids in the hemp oil.

-The ratio in ω-6/ω-3 is most represented by linoleic acid (LA; 18:2, n-6) and α-linolenic acid (ALA; 18:3, n-3). As suggested by reviewer, the authors changed in “HS oil is a rare fount of nourishment for human diet, in particular for vegetarians, because of their unique reported ratio in ω-6/ω-3, most represented by linoleic acid (LA; 18:2, n-6) and α-linolenic acid (ALA; 18:3, n-3)”.

  1. Line 220-255: Please add some discussion for the results of chlorophylls and carotenoids (the meaning of their concentrations, ratios, global regulations, etc.).

-As suggested by reviewer 1, the authors amply this section adding more details.

  1. Line 251: What is chlorophylls/carotenoids ratio for? Any regulation for this?

-The chlorophylls/carotenoids ratio is able to indicate phenology and physiology of plants. Different varieties from different sources, harvested in a similar ripeness state, possess different chlorophyll and carotenoid pigment profile and content. Different chlorophyll a/b content and chlorophyll/carotenoid ratios caused by stress, damage, senescence, impact the normal course of plant biological processes. Even though there is not a restrict regulation on the ratio between chlorophylls and carotenoids, it tends to remain more or less constant whatever the variety of sources, in a concentration range of 2.5 to 3.7 mg total chlorophyll/mg total carotenoid.

  1. Line 260-268: Please add more discussion for the results of cannabidiolic acid (meaning, regulations, etc.).

-As suggested by reviewer 1, the authors added some details to the discussion of cannabidiolic acid.

  1. Line 359-370: What is the meaning of PCA analysis in this study? How can the results of PCA contribute to the quality control of hemp seed oil? I didn't get the point of grouping of oil samples on the PCA score scatter plot. The results need more discussion as well.

-As also suggested by reviewer 2, the authors decide to remove the multivariate principal components analysis (PCA) because of the limited number of samples not allow to obtain better visualization of the output data and better visualization of the relationship between variables.

  1. Line 392: I am wondering if there were any biological and/or technical replicates for each sample?

-Replicates are referred to as technical replications.

  1. Line 459-460: Which method was used for LC-MS quantification (calibration curve, etc.)? On top of that, how did author correct deviations from extraction and MS ionization (internal standards, etc.)?

-A linear calibration curve of eight points was constructed covering a concentration range from 0.5 to 500 mg/kg. The proposed analytical method was in-house validated (in terms of linearity, inter- and intraday precision, accuracy, sensitivity, and selectivity) in a previous study. Quantification was performed comparing the retention time of samples’ peak with standard peak.

The authors thank reviewer for evaluating the manuscript.

Reviewer 2 Report

The work contains important research results, but the authors also made methodological errors.

1)The abstract of the work should be short and contain the purpose of the work, short work methodology and conclusions. The abstract in the included work does not contain all the elements and, above all, does not contain the results.

2)The introduction should relate to the topic of work and include a research hypothesis.

3)After the chapter "Introduction" you should choose the chapter "Material and methods", the second chapter to "Results and discussion" and "Conclusion" as chapter 3. Therefore, you should change the order of chapters.

4)In the methodology, failure to check the place of sampling, which is important from the point of view of the commodities science.

5)Performing statistical calculations with only 3 repetitions for each objects provokes tests (too little variability), moreover:

  a.)although the authors write that the differences between the mean values were taken using t-Student's test, at the significance level of 0.05, these values ​​are not included in the table and the results are not discussed according to with statistically significant differences;

  b.)was calculated mean arithmetic and standard deviation, the SPSS program should also calculate and describe other descriptive indicators of statistics, such as: median, skewness, kurtosis, as well as coefficient of variation, which better than other characteristics variability descripting indicators;

  c.)the correlation coefficient r (from the sample) is a summary of the overall q correlation results and is therefore already mistake biased. The correlation coefficient is a statistic and therefore should be treated as a random variable;

  e.)conducting multivariate principal components analysis (PCA) was aimed at obtaining better visualization of the output data and better visualization of the relationship between variables, but with such a small number of data it should not be performed.

6)The "Results and Discussion" section should be discussed separately.

7)Conclusions should be summarizing and generalizing, and should not contain research details or vague suggestions.

Author Response

Response to reviewers

Manuscript ID: molecules-883344

Title: Analyzing commercial hemp seed oil from Mediterranean basin: how far have we got in its chemical goodness?

Reviewer 2

Comments and Suggestions for Authors

The work contains important research results, but the authors also made methodological errors.

1)The abstract of the work should be short and contain the purpose of the work, short work methodology and conclusions. The abstract in the included work does not contain all the elements and, above all, does not contain the results.

- As suggested by reviewer 2, the authors rewrite and reorganize abstract including the main results: “Although hemp seed (HS) oil is characterized by more than 80% polyunsaturated fatty acids (PUFAs), a very high ω-6 to ω-3 ratio is not a popular commodity. The aim of this work was to provide useful data about the bioactive components and cannabidiolic acid content in thirteen different commercial hemp seed oils. The investigated HS oils showed a good ω-6/ω-3 ratio, ranging from 1.71 to 2.27, massively differed in their chlorophylls (0.041-2.64 µg/g) and carotenoids contents (0.29-1.73 µg/g), as well as in total phenols (22.1-160.8 mg GAE/g) and tocopherols (3.47-13.25 mg/100g). Since the high content of PUFAs in HS oils, photo-oxidative stability was investigated by determining the Thiobarbituric Acid Reactive Substances (TBARS) assay and extinction coefficient K232 and K270 after the photo-oxidative test. The percentage of increase in K232 and K270 ranged from 1.2 to 8.5% and from 3.7 to 26.0%, respectively, meaning good oxidative stability, but had an oxidative behavior by TBARS, showed a 1.5- to 2.5-fold increase when compared to the initial values. Therefore, the diversity in bioactive compounds in HS oils, and their high nutritional value, suggesting the need for a disciplinary booklet that well defines agronomic and post-harvest management conditions for achieving a good food objective.”

2) The introduction should relate to the topic of work and include a research hypothesis.

- As suggested by reviewer 2, the authors added the missing information in the introduction section

3) After the chapter "Introduction" you should choose the chapter "Material and methods", the second chapter to "Results and discussion" and "Conclusion" as chapter 3. Therefore, you should change the order of chapters.

- The authors prepared the manuscript follow the Molecules Microsoft Word template file available on the official website: https://www.mdpi.com/journal/molecules/instructions

4)In the methodology, failure to check the place of sampling, which is important from the point of view of the commodities science.

- As suggested by reviewer 2, the authors added the place of sampling in the methodology.

5)Performing statistical calculations with only 3 repetitions for each objects provokes tests (too little variability), moreover:

  1. a) Although the authors write that the differences between the mean values were taken using t-Student's test, at the significance level of 0.05, these values ​​are not included in the table and the results are not discussed according to with statistically significant differences;

- As suggested by reviewer 2, the authors included the significance level in the table and discussed the results according to the statistically significant differences.

  1. b) was calculated mean arithmetic and standard deviation, the SPSS program should also calculate and describe other descriptive indicators of statistics, such as: median, skewness, kurtosis, as well as coefficient of variation, which better than other characteristics variability descripting indicators;

- As suggested by reviewer 2, the authors added other indicators of statistics including measures of skewness and kurtosis.

  1. c) The correlation coefficient r (from the sample) is a summary of the overall q correlation results and is therefore already mistake biased. The correlation coefficient is a statistic and therefore should be treated as a random variable;

-As rightly suggested by reviewer 2, the authors delete the mistake in the statistical analysis section.

  1. e) Conducting multivariate principal components analysis (PCA) was aimed at obtaining better visualization of the output data and better visualization of the relationship between variables, but with such a small number of data it should not be performed.

- As suggested by reviewer 2, the authors removed the multivariate principal components analysis (PCA) in the manuscript.

6) The "Results and Discussion" section should be discussed separately.

As reported in the Instructions for Authors, in the manuscript the results can be combined with discussions. The authors decided to combine the "Results and Discussion" section because they believe that the reader may understand it easily.

7) Conclusions should be summarizing and generalizing, and should not contain research details or vague suggestions.

As suggested by reviewer 2, the authors changed the conclusions as: " Nowadays, consumers are constantly looking for useful natural products to supplement the human diet in order to prevent or treat human diseases. Despite some reticence towards this matrix, hemp seed oil has been heavily studied in recent decades. So, several studies reported the beneficial effect deriving from oil extracted from hemp seed. Indeed, although the use of hemp seed in Italy, as part of human diet as such it is or as its by-product (e.g. oil), has been legislated since 2009 [79], little is known about its chemical composition and the variability of this latter. This can have several primary consequences, among which the marketing of highly dissimilar products from an analytical-quantitative point of view. The uniqueness of the hemp seed oil product must be the subject/object of greater attention. Data herein reported showed that investigated HS oils markedly differ in the content of some bioactive compounds, suggesting the need for a disciplinary booklet that well defines agronomic and post-harvest management conditions for achieving a good food objective. "

The authors thank reviewer for evaluating the manuscript.

Reviewer 3 Report

Interesting paper with several analysis to characterize hemp oil. Suitable for publication after some minor changes. find my comments below:

Title: "Mediterranean basin" does not reflect the actual sampling as it does not include representative samples from it but for a few regions of Italy.

In the entire manuscript, check significant digits as they are not consistent throughout the paper. 

Some additional discussion on the results and the multivariate analysis should be included.

As a recommendation, if the authors aim for more studies on hemp oil, which still has potential, use of state-of-the art analytical methods is highly encouraged.

Author Response

Response to reviewers

Manuscript ID: molecules-883344

Title: Analyzing commercial hemp seed oil from Mediterranean basin: how far have we got in its chemical goodness?

Reviewer 3

Comments and Suggestions for Authors

Interesting paper with several analysis to characterize hemp oil. Suitable for publication after some minor changes. find my comments below:

1) Title: "Mediterranean basin" does not reflect the actual sampling as it does not include representative samples from it but for a few regions of Italy. In the entire manuscript, check significant digits as they are not consistent throughout the paper. 

- As rightly suggested by reviewer 3, the authors removed the term “Mediterranean basin” and corrected the significant digits in the text.

2) Some additional discussion on the results and the multivariate analysis should be included.

-As also suggested by reviewer 2, the authors decide to remove the multivariate principal components analysis (PCA) because of the limited number of samples not allow to obtain better visualization of the output data and better visualization of the relationship between variables.

3) As a recommendation, if the authors aim for more studies on hemp oil, which still has potential, use of state-of-the art analytical methods is highly encouraged.

- The authors thank reviewer 3 for their valuable advice.

The authors thank the reviewer for evaluating the manuscript.

Round 2

Reviewer 1 Report

The manuscript was revised correctly according to reviewers' comments.

Reviewer 2 Report

Work after the improvement does not raise any objections. The introduction was supplemented as well as the missing descriptive statistics calculations. The description of unauthorized calculations has also been deleted. The requests have been corrected.